# Peer-Education as a Tool to Educate on Antibiotics, Resistance and Use in 16–18-Year-Olds: A Feasibility Study

**DOI:** 10.3390/antibiotics9040146

**Published:** 2020-03-30

**Authors:** Cliodna A. M. McNulty, Rowshonara B. Syeda, Carla L. Brown, C. Verity Bennett, Behnaz Schofield, David G. Allison, Neville Q. Verlander, Nick Francis

**Affiliations:** 1Primary Care and Interventions Unit, Public Health England, Gloucester GL1 1DQ, UK; rowshonara.syeda@phe.gov.uk (R.B.S.); carlalbrown@gamedrlimited.com (C.L.B.); 2Division of Population Medicine, Cardiff University, Cardiff CF14 4XN, UK; BennettCV@cardiff.ac.uk; 3Faculty of Health and Applied Sciences, University of West of England, Bristol BS16 1QY, UK; behnaz.schofield@uwe.ac.uk; 4Division of Pharmacy & Optometry, University of Manchester, Manchester M13 9PT, UK; David.Allison@manchester.ac.uk; 5Statistics, Modelling and Economics Department, Public Health England, London NW9 5EQ, UK; Neville.Verlander@phe.gov.uk; 6School of Primary Care, Population Sciences and Medical Education, University of Southampton, Southampton SO17 1BJ, UK; Nick.Francis@soton.ac.uk

**Keywords:** peer education, biology, students, antibiotics, antibiotic resistance, health education

## Abstract

Peer education (PE) interventions may help improve knowledge and appropriate use of antibiotics in young adults. In this feasibility study, health-care students were trained to educate 16–18 years old biology students, who then educated their non-biology peers, using e-Bug antibiotic lessons. Knowledge was assessed by questionnaires, and antibiotic use by questionnaire, SMS messaging and GP record searches. Five of 17 schools approached participated (3 PE and 2 control (usual lessons)). 59% (10/17) of university students and 28% (15/54) of biology students volunteered as peer-educators. PE was well-received; 30% (38/127) intervention students and 55% (66/120) control students completed all questionnaires. Antibiotic use from GP medical records (54/136, 40% of students’ data available), student SMS (69/136, 51% replied) and questionnaire (109/136, 80% completed) data showed good agreement between GP and SMS (kappa = 0.72), but poor agreement between GP and questionnaires (kappa = 0.06). Median knowledge scores were higher post-intervention, with greater improvement for non-biology students. Delivering and evaluating e-Bug PE is feasible with supportive school staff. Single tiered PE by university students may be easier to regulate and manage due to time constraints on school students. SMS collection of antibiotic data is easier and has similar accuracy to GP data.

## 1. Introduction

Antimicrobial resistance (AMR) is a growing threat to global health [1]. Public education helps to improve antibiotic use; this includes public health initiatives such as World Antibiotic Awareness Week, local public campaigns and educating young people [1]. Antibiotics are a commonly prescribed childhood medicine [2]. Educational resources targeting hygiene for young people should help to reduce the rate of infections and therefore antibiotic use. Young people aged 15–24 years have a lower understanding of conditions that are effectively treated by antibiotics compared to other age groups [3]. Moreover, targeting education to 16–18-year olds is important as this group are beginning to take care of their health and self-care for infections such as colds and flu [4]. e-Bug, operated by Public Health England (PHE), is an educational resource for teaching young people of all ages on microbes, the spread, treatment and prevention of infection and prudent antibiotic use [5]; e-Bug includes peer-educational resources [6].

Peer-education (PE) is a popular learning tool that has been successfully used for teaching a range of health behaviours including sexual health [7]; smoking [8], diet and physical activity [9,10]. e-Bug PE resources on microbes, hygiene and antibiotics, used in a science-show based format [6] significantly increased antibiotic knowledge in 12–13-year-old peer educators and the 9–11-year-old peers that they taught, and also improved behaviour, confidence and communication skills [6]. Although PE has been used by university medical students [11] and was shown to reduce self-reported antibiotic use for colds and flu [12], a NICE systematic review found no PE studies that measured reductions in antibiotic prescriptions [13].

This study aimed to investigate the feasibility of university healthcare student-led antibiotic PE lessons, delivered to 16–18-year-old biology students in schools; and the subsequent delivery of these lessons by the biology students to their 16-18-year-old non-biology peers. The study also tested the feasibility of measuring reported student antibiotic use through questionnaires and text messaging, compared to the gold standard of GP record searches.

## 2. Results

Five of 17 (29%) schools approached participated. In Cardiff, 14 secondary schools were invited to participate by email and the first three schools that agreed to take part were included in the study (two randomised to intervention, one randomised to control). In Manchester, two out of three secondary schools approached agreed to participate (one randomised to intervention, one to control). Ten of 17 (59%) university students approached agreed to volunteer as peer educators (PEds). In Cardiff, six of 13 medical students taking an intercalated BSc course agreed to participate as PEds. In Manchester, all four pharmacy students that were approached randomly agreed to participate as PEds (Figure 1a,b). 

### 2.1. Intervention Schools

Fifteen of 54 biology students volunteered to be PEds after an antibiotic lesson. In Cardiff, 33 A-level biology students (26 in intervention school A and 7 in intervention school B) attended PE lessons (one lesson in each school). In intervention school A, 9 (35%) of the 26 biology students went on to be trained as PEds and these PEds then taught one lesson to 11 non-biology students. In intervention school B, 6 (86%) of the 7 biology students went on to be trained as PEds. This equates to 45% (15/33) of all biology students receiving PE in Cardiff schools volunteering to become PEds themselves. Cardiff PEds worked in groups of 3 to teach a further 6 lessons (127 students) with their non-biology peers. Some of these lessons in school B were experienced as challenging, as PEds sometimes had inadequate space to teach, were provided with little or no support in controlling class behaviour, and had students with poor English language skills or who displayed behavioural problems. This was so bad in one lesson of 24 students (lesson 7) that it was impossible to gather reliable questionnaire data and therefore no student questionnaires were completed for this lesson. Thirty percent of non-biology students completed all three questionnaires in Cardiff (38/127) range 0–75% in school B, 81–100% in school A). The number of students attending, participating, and completing follow-up questionnaires from each lesson are given in Table 1. In Manchester, only one tier of PE was undertaken, in which four university students delivered the antibiotics lesson to one group of 21 biology students. These students did not carry out any further PE themselves due to the study being at the same time as exams. In the Manchester intervention school, 21 (100%) students completed the pre, 19 (90%) completed the post, and 15 (71%) completed the follow-up (three-month) questionnaires. 

### 2.2. Control Schools in Cardiff and Manchester 

In both control schools, 16-18-year-old biology students continued their usual biology course, and the researcher visited the school at a convenient time for the teacher. Schools were offered the antibiotics lesson in the future, but none took up the offer. In the Cardiff control school, 95 biology students were given information about the study and invited to attend lunchtime sessions to complete consent forms and questionnaires over four consecutive days. Fifty out of 95 (53%) students attended; 50 (100%) completed the pre-questionnaire, and 45 (90%) the three-month follow-up questionnaire. In the Manchester control group: 18/25 (72%) completed the pre-questionnaire and 21/25 (88%) completed the three-month questionnaire during normal lesson time.

### 2.3. Questionnaire Knowledge Scores 

Although Cardiff intervention school B had lower questionnaire scores than school A (Figure 2a), they showed a similar pattern of improvement post-intervention, with a small fall in knowledge at follow-up (School B: pre- median = 6, IQR = 3–8; postmedian = 8, IQR = 5–12; follow-up median = 7, IQR = 4–11). The range of knowledge scores of Cardiff intervention students (a mix of biology and non-biology students) had a bimodal distribution with a smaller percentage in each school attaining high knowledge scores. Intervention school A students which comprised a higher percentage of biology students (70% in A vs. 8% in school B) and served a less deprived area answered more questions correctly and had a narrower range of correct answers. In the Cardiff control school (Figure 3) these biology students scored similarly to intervention school A (70% biology) at pre (median = 12, IQR = 10–13), with one outlier scoring 5, at follow-up (median = 11, IQR = 9–13). 

The students in the Manchester intervention school, all studied biology, and generally scored higher at pre (median = 13, IQR = 11–14), post (median = 13, IQR = 12–14), and follow-up (median = 14, IQR = 11–15) compared to the Manchester control school students, who were a mix of biology and non-biology students (Figure 2b). Biology student knowledge scores were slightly higher post PE lesson, and there was no significant fall in knowledge after three months. Manchester control students (a mix of biology and non-biology students) also had lower knowledge scores than Cardiff control students (all biology), pre (Manchester median = 8, IQR = 5–12 versus Cardiff median 12, IQR 10–13), and three-month follow-up (Manchester median 8, IQR 7–12 versus Cardiff median 11, IQR = 9–13). Logistic regression showed a significant increase in knowledge after the intervention for five of the 16 questions: how to treat coughs and colds (question 2), being able to identify antibiotic names (question 3), that antibiotics treat bacteria but not colds and viruses, (question 5 and 6) that antibiotics kill good and bad bacteria in the gut (question 7), and that you should not use leftover antibiotics (question 16). Question 16 also led to a significant improvement in knowledge gain for non-biology students (*p* < 0.001). 

### 2.4. Assessing Antibiotic Use

In Cardiff, 64/136 students provided consent for the research team to contact their general practitioner and provided recognisable GP surgery data. As this study was designed to test the feasibility and there were a low number of participants, responses for control and intervention students were combined. The 64 students were registered with 34 different GP practices, with one to six students registered at each GP practice. Thirty-three of the 34 GP practices agreed to provide data (one practice with six students declined) and 31 practices were contacted. Data were not available for four of the 58 students (one had left the surgery and three were not registered with the surgery name given). Of the 54 students with antibiotic prescribing data, 7 (13%; 5 controls, 2 intervention) had been prescribed an antibiotic during the follow-up period and 47 (87%; 24 control, 23 intervention) had not been prescribed an antibiotic. 

Consent to receive text messages asking about antibiotic use, and a valid phone number was given by 98 students (72% of 136 students that completed questionnaires). 69/98 students (70%) responded to at least one text message over three months. 53/98 (54%) responded to questions every month for three months. Monthly response rate decreased over the three months from 66% (65/98) in January 2018 to 56% (55/98) in March 2018. Both GP and SMS data were available for 35 students, and 22 students who responded to the SMS every month also had data from their GP practice. Nearly all students (93% 54/58) responding to the first text message without reminders in February, did so within 24 hours. Most responding students (91%, 59/65) answered “No” to question one; 88% (56/64) in February; 95% (52/55) in March. 

There was 91% agreement of antibiotic use reported by text messaging, by students responding to any text compared to practice search of GP data (Table 2, kappa = 0.72), and 86% agreement when restricted to those students who responded to all 3 texts (Table 2. kappa = 0.67) One student reported taking an antibiotic but indicated that this was a long-term treatment for a skin condition. Excluding this student increases agreement to 94%, with a kappa of 0.80. Two students reported that they had received an antibiotic from their GP surgery, however, the surgery reported that they had not been prescribed an antibiotic. 

109/136 students from the Cardiff schools (64 intervention and 45 control) provided data on antibiotic use on their three-month follow-up questionnaires. 29 (27%) reported taking one or more courses during the follow-up period. Three-month questionnaire data and GP prescribing data were available for 25 students, and the agreement was very poor with a kappa of 0.06. Eight students reported taking antibiotics, but their GP records showed no evidence of being prescribed antibiotics. Agreement between the three-month questionnaire data and the SMS data was better. 20 students had questionnaire data and full SMS data, and there was agreement for 75%, giving a kappa of 0.39.

## 3. Discussion

### 3.1. Main Findings 

In this feasibility study, PE using e-Bug resources in secondary schools, we were able to successfully recruit five schools (three intervention and two control) who managed to recruit 181 intervention students (160 in Cardiff and 21 in Manchester) to attend a peer education lessons. We were also able to recruit medical and pharmacy university students who were enthusiastic and delivered e-Bug lessons to biology students in the 3 intervention schools. There were challenges in recruiting schools and students, with schools being more willing and able to participate if the study fitted into the school timetable, with recruitment and delivery outside of exam periods. Biology students, whether intervention or control, had higher knowledge scores compared to non-biology students. The pre-lesson knowledge scores of biology students suggest that they had the knowledge to give PE lessons and this increased post-lesson; but their educational skills were less refined. Using biology students to deliver lessons without teacher supervision to non-biology students was a problem in some classes, due to participants poor behaviour and language barriers. Due to study timing near school exams, school student peer to peer education was not feasible in Manchester.

We found some evidence for improved knowledge scores for some questions and in some groups after delivery of the lesson. However, this study was not designed to assess the effects of the intervention on knowledge and was underpowered for this outcome. Antibiotic data collection by student reporting via SMS and questionnaire were more efficient and feasible approaches than collecting data from a multiplicity of GP practices. There was good agreement between GP antibiotic data and SMS data, but the poor agreement between the GP data and three-month questionnaire data, and only moderate agreement between the questionnaire data and SMS data, but it is unclear which approach best reflects actual use. 

### 3.2. Strengths and Limitations

The study was conducted in two universities, schools were invited in random order and participating schools had a range of cultural backgrounds and educational needs, indicating that the model is likely to be transferable and feasible in many different school and university settings within England and Wales if undertaken outside school exam times, and with teacher supervision, if student disruption is likely. To our knowledge, this is the first time that text messaging has been compared to standard searches of GP records for 16–18-year olds, with consistent results. 

Students and researchers reported that the consent process and questionnaire completion reduced available lesson time. In one Welsh school, teachers only offered voluntary PE lessons during the lunch break. This demonstrates that flexibility in lesson delivery is key to ensuring PE reaches schools that are interested but need to adhere to the usual timetable, yet still ensuring that all learning outcomes are met. 

The control school in Cardiff and intervention school in Manchester chose to involve only biology students in the study as it was easier to complete the antibiotic-related questionnaire collection within a biology subject. Biology students all had higher baseline knowledge about the subject matter of antibiotics and consequently, there was less opportunity for improvement in knowledge scores. In line with ethics approval, the questionnaire results were anonymised and in schools where biology and non-biology students received PE, results cannot be differentiated. In future evaluations, ethical approval and consent should be obtained to differentiate peer educators and different student groups, or questionnaire data should capture this information. As this was a pragmatic feasibility study and schools could withdraw at any time there was drop-out of students. In Manchester, there was only one tier of PE with the university students delivering the lesson to 16-18-year-old biology students. 

A Public Health England national antibiotic resistance campaign was launched during the time of the study between October 2017–January 2018, however, as participants were not asked about whether they had seen the campaign or video, it is not possible to ascertain if this influenced the results.

### 3.3. Other Research in this Area 

Previous research demonstrates the feasibility of this type of teaching as there are benefits aside from knowledge increase [7,10]. In a study using a similar model, university students trained 15–16-year-old PEds on ways to improve diet, physical activity and screen time habits [9]. These 15–16-year olds delivered the intervention to 12–13-year-olds, leading to improvements in these variables two weeks after the intervention. 

Another study involving 82 students in Portugal, measured knowledge change in 14–16-year-old students [14] before and two months after a 90-minute presentation covering antibiotics and antibiotic resistance; results showed knowledge of correct use of antibiotics against bacterial infection increased from 43% to 76% (*p* < 0.05) and knowledge of the risks of incorrect use increased from 48% to 74%. In the present study, antibiotics PE was targeted towards older knowledgeable biology students of 16-18 years, who then taught similar-aged non-biology peers. 

Challenges such as time constraints, pressures to follow the school curriculum, lack of support for integrating PE or appropriate training to facilitate implementation, and clashes with exams and study leave, have been identified in other studies as well [15]. Reducing these barriers could increase successful uptake in the future. 

Other studies investigating knowledge, attitudes and behaviour towards antibiotic use amongst university students have identified poor knowledge on when antibiotics should be used and how to take them correctly [16]. Having good knowledge does not always transfer to actual behaviour, leading to an intention-behaviour gap [17]. Implementation intentions have been shown to be effective in improving health behaviours such as a healthy diet [18]. For this reason, it will be useful to add a goal-setting and action-planning section to the lesson plan. Our PE lesson included discussion about a self-care leaflet and where to seek help, however, the time available for this was very short due to questionnaire completion. The texting process to collect antibiotic use data will be an excellent way to determine whether PE around antibiotic use also leads to behaviour change. 

### 3.4. Implications for Implementing Peer-Education and Future Work

School peer educators were given two hours of peer educator training and encouraged to practice delivery in their own time. However, differing timetables made it difficult for students to find mutually convenient times to practice. Furthermore, If PE training were to be undertaken by university students in a setting with time constraints for both students and teachers, such as in place of a lesson or a lunchtime period, this may be easier to manage and regulate. PE resources should include guidance on classroom management techniques and different ways to present information. We found that for some of the knowledge questions, all students had a high level of baseline knowledge. We suggest that questions answered correctly by more than 80% of students before the lesson should be removed. This would result in our questionnaire being reduced from 16 to 10 items. 

The results indicated there was less knowledge improvement post PE about antibiotic-resistant bacteria, its carriage and spread; this may be a difficult concept for some students to understand fully and teach others about, especially for non-biology students. During the lesson, the topics ‘gut bacteria’ and ‘resistant bacteria in healthy people’ were covered last, so this activity may have been rushed. In future implementation, the full lesson period should be used for teaching rather than consent and questionnaire completion. The information about bacterial and viral infections causing similar symptoms may have been misunderstood by English biology students as they were less likely to get this right after the PE lesson; consideration should be given to making this clearer in future studies. Future studies could also place more emphasis on behaviours, as well as learning outcomes to determine how behaviours may positively change with PE on antibiotics. 

Correct and incorrect use of antibiotics, including not sharing antibiotics or keeping leftovers, are important messages students need to learn about, and this would encourage self-care and decrease inappropriate antibiotic use. These concepts featured throughout the lesson, and led to greater knowledge gain, particularly in question 16 for non-biology students (*p* < 0.001, 32% to 76%). We suggest that the PE lessons for 16-18-year-old students should cover the same activities but go into more depth to provide students with greater knowledge that matches better to the biology curriculum (Box 1).

Box 1Lessons learnt.
Biology students had the knowledge to deliver antibiotic lessons but lacked the teaching skills or support from teachers to facilitate education in some classes. Consent should be obtained outside the peer education lesson delivery to allow sufficient time to complete the lesson activities and action planning.Peer education needs to be arranged to avoid exam times of university and school students.Future questionnaires should be able to differentiate students by biology, non-biology and peer-educator status; ethical approval may be needed to do this.Knowledge questionnaires can be shortened by removing questions with a high correct response rate in the pre-intervention questionnaire.Evaluation questionnaires should be modified to collect students antibiotic use intentions before and after lessons, and knowledge of any other campaigns that may have influenced their behaviour.Students attending a single school can be registered at a wide range of GP practices, which makes antibiotic use data collection very challenging.SMS messaging was found to be a feasible and valid approach to collecting antibiotic use data, with good response rates from students and good agreement with GP antibiotic prescribing data obtained by note searching.Although student self-reported classroom antibiotic use questionnaires gave the most data, there was a very poor correlation with GP antibiotic data, therefore this method should not be used in future evaluations to measure student antibiotic use.Single tier peer-education by university healthcare students to 16-18-year-old students may be more feasible than two-tier peer-education using both university and student educators.


## 4. Materials and Methods 

### 4.1. Study Design 

A feasibility study of a peer-education evaluation using a cluster Randomised Control Trial design. 

### 4.2. Objectives

To determine the feasibility of university, school and student recruitment, consent procedures in schools, pre, post and follow-up questionnaire completion, and student antibiotic use data collection. 

### 4.3. Setting

The project was undertaken in a convenience sample of secondary schools close to the two participating universities (Cardiff University and the University of Manchester). 

### 4.4. University Student Recruitment

Using convenience sampling, third-year pharmacy students from the University of Manchester, and medical students from Cardiff University undertaking an intercalated BSc, were invited to be trained as PEds. All university students that volunteered to be PEds and deliver an e-Bug lesson on antibiotics within the school setting were provided with training and offered a certificate for their portfolios, but were not provided with any other remuneration. 

### 4.5. School Recruitment

A convenience sample of schools in Cardiff and Manchester, close to the two universities were selected to take part, through invitation letters, email and follow-up telephone calls. Participating schools were randomly assigned to either intervention or control school using random numbers in Excel. In Cardiff, student consent for all aspects of data collection were obtained by the researcher immediately before PE lessons. In Manchester, parents were sent an opt-out form in advance of the study being carried out. In each area (Cardiff and Manchester) we aimed to recruit two intervention schools, each of which would recruit 10 student PEds who would deliver lessons to 100 students. With attrition rates, we expected that 75 students would complete questionnaire data in intervention schools. For this feasibility study, we aimed to recruit one control school in each area and 75 students to complete questionnaires. We planned to offer control schools antibiotic lessons after the antibiotic and knowledge data collection was completed. 

In Cardiff, six university PEds delivered the lesson to 16-18-year-old biology students in each school in groups of three to four; while in Manchester four university PEds delivered the lesson to 16-18-year-old biology students. In each school, some biology students then volunteered to be trained as PEds and deliver the lesson to their peers, either during a timetabled lesson (in Cardiff) or during the lunch break (in Manchester). 

### 4.6. Peer Educator Recruitment and Training

University student PEds were trained by a member of the research team and the 16-18-year-old biology students were trained by either a member of the research team or the university students. All PEs were given the e-Bug PE lesson plan, including a PowerPoint presentation explaining antibiotics and antibiotic resistance, interactive activities and group discussion activities. All PEds were provided with a checklist of keywords and learning outcomes to cover during the lesson and were encouraged to meet with each other before the delivery of the intervention to practice lesson delivery and content. University students delivered the PE antibiotics lesson to 16-18-year-old biology students. Mixed-gender biology students who received the lesson then volunteered to be school PEds themselves. PE lessons in intervention schools were also observed by a researcher in each region. 

### 4.7. Materials 

A common e-Bug antibiotics PE lesson plan was developed for use in all university-led and 16-18-year-old led antibiotics PE lessons, to ensure consistent messages and learning. This included background information for the educator, learning outcomes, a PowerPoint presentation, interactive demonstrations, scenarios about young people with common infections, and right or wrong statements about antibiotic use for students to discuss. PEds were also provided with answer sheets for scenarios and right or wrong statements to facilitate discussion during the lessons. 

### 4.8. Data Collection 

#### 4.8.1. Feasibility Measures

The researchers collected data on university and student uptake of PE, consent for participating in PE lessons; consent to receive text messages from researchers, and/or database search for their GP records on antibiotic use.

#### 4.8.2. Evaluating Knowledge on Antibiotic Use 

A knowledge questionnaire was developed based on previous e-Bug evaluations and discussions between the research group which consisted of a GP, microbiologist, epidemiologist and researcher. The questionnaire covered the key learning outcomes for the lesson and included questions on microbes, antibiotics, antibiotic resistance and typical young person infection scenarios depicting correct and incorrect antibiotic usage. The questionnaire was piloted with sixteen 16-17-year-old biology school students in Gloucestershire and modified iteratively to produce a 16-item multiple-choice questionnaire. (Appendix A: The following are available online at http://www.mdpi.com/2079-6382/9/4/146/s1, Appendix A: Antibiotics lesson plan, Appendix A: Pre-knowledge questionnaire, Appendix A: Immediately post-knowledge questionnaire; Appendix A: 3-month follow-up questionnaire).

Intervention school students were asked to complete the same questionnaire immediately before the lesson (pre), immediately after (post), and three months later (follow-up). Control school students completed pre- and three-month follow-up questionnaires. 

#### 4.8.3. Antibiotics Use Data Via Questionnaires, Text Messaging and GP Record Data Collection (Cardiff)

At the beginning of each e-Bug PE lesson (in intervention schools), or pre- questionnaire session (in control schools), students were invited to provide their consent and mobile telephone number to take part in a text message survey, and to provide their consent to access GP records and provide a surgery name. 

Reported antibiotic usage data was collected in three-month follow-up questionnaires, by SMS (text messaging) and from general practice record searches from all consenting students in Cardiff in both control and intervention schools. Using the SMS system “Textanywhere” (https://www.textanywhere.net/), texts were sent every month for three consecutive months. The survey asked one or two questions (according to the first response) about antibiotics (Figure 3). If no response was given to the first question, a reminder text message was sent one week later. The proportion of pupils responding to the first text message after 24 hours was also recorded.

GP practice records for consenting students were searched for antibiotic use. Health Care Assistants at each consenting GP surgery searched each student GP record for antibiotic prescriptions over the same 3-month period over which the text messages were sent, and questionnaire surveys were undertaken. 

### 4.9. Data Analysis

Questionnaire data were entered into EpiData version 4.2 software, with all intervention pre, post and follow-up questionnaires matched by a unique student ID number to ensure anonymity. Data were exported to Excel and analysed descriptively, including average scores and totals for questions answered correctly. Twenty per cent of the data were also double entered. Only two discrepancies were identified, and these entry errors were corrected. The database was shared with PHE statistician N.Q.V and exported into STATA version 15.1, in which data manipulation and non-exploratory statistical analysis for knowledge change were performed by N.Q.V. Results are presented as proportions, medians and ranges, and questionnaire scores are displayed as box and whisker plots. Logistic regression examined possible effects of the intervention on student knowledge. Each question was analysed separately with question response (correct/not correct) as the outcome. The logistic regression model consisted of the main effects of period (pre-intervention, post-intervention and 3-month follow-up), country (two categories for England and Wales) and their interaction. This was then simplified by omitting the interaction if it was not statistically significant. 

We used Cohen’s kappa statistic to calculate agreement between different approaches to collecting data on antibiotic use (general practice medical records, self-report by SMS messaging, and self-report through completion of the follow-up questionnaire), matching each student SMS data to questionnaire data and GP record data. As this was a feasibility study, control and intervention student data were analysed together.

Ethical Considerations: The study gained local ethical approval from PHE Research Ethics Governance Group and NHS Ethics approval (REC reference: 17/LO/1512) for GP record data collection in Cardiff. 

## 5. Conclusions

Our study demonstrates that cascading the activities from university health students to 16-18-year-old biology students, and then to their non-biology year group, is feasible and may increase antibiotic knowledge. Therefore, this approach should be considered with health care students and schools. It would be beneficial to conduct a larger study involving more schools, teachers and universities, and confirm the benefits of this approach, using SMS data collection for assessing antibiotic use. Future studies should consider the creation of two separate lesson plans, to firstly give a comprehensive overview of what antibiotics are and what causes antibiotic resistance, and second go into greater detail about how to prevent resistance through self-care and appropriate antibiotic use. This would also allow peer educators to prioritise what is taught in each lesson and educators would be able to spend more time teaching complex concepts in greater depth. In future evaluations, school and student participation could be improved by timing the study outside exam and coursework periods and providing incentives. The evaluation survey should be shortened thereby removing non-discriminatory questions. 

## Figures and Tables

**Figure 1 antibiotics-09-00146-f001:**
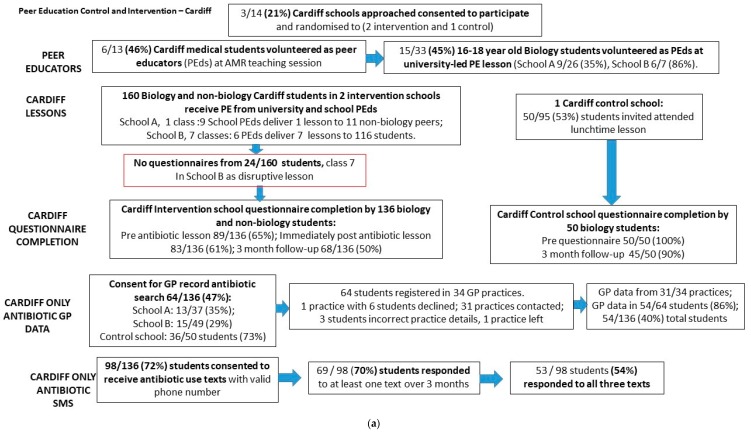
(**a**) Overview of intervention and the number of complete questionnaires and text messaging data in Cardiff. (**b**) Overview of intervention and the number of completed questionnaires in Manchester.

**Figure 2 antibiotics-09-00146-f002:**
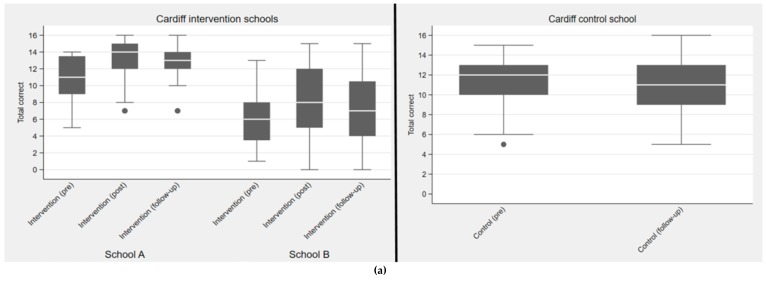
(**a**) Box and whisker plots for Cardiff intervention and control schools for pre, post (intervention only) and a 3-month follow-up questionnaire. (**b**) Box and whisker plots for Manchester intervention and control school. Boxes show interquartile range (IQR) with the median as a horizontal line, whiskers include data points within 1.5× IQR of nearest quartile.

**Figure 3 antibiotics-09-00146-f003:**
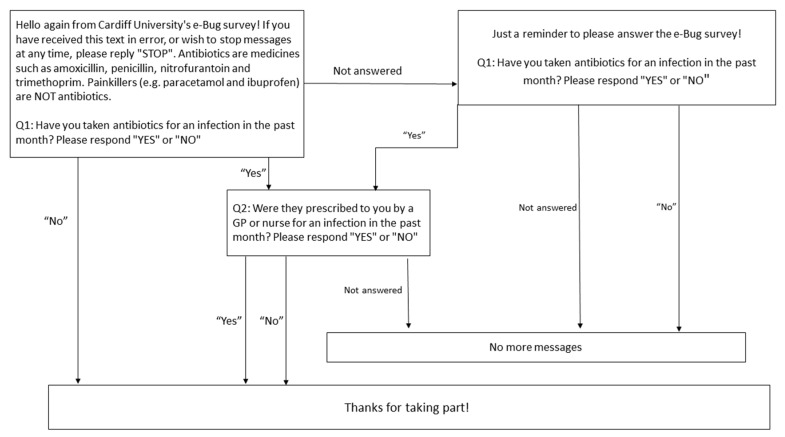
Text message survey question and answer structure for each monthly survey.

**Table 1 antibiotics-09-00146-t001:** School students in Cardiff and Manchester intervention and control schools invited to and attending lessons, and completing pre- and immediately post and 3-month post-lesson questionnaires (*N* = 301).

Intervention School	Class	Students Attending Lesson	Students Completing Pre Lesson Questionnaire		Students Completing Post- PE Lesson Questionnaire		Students Completing 3-Month Questionnaire	
		Number (N)	*N*	%	*N*	%	*N*	%
**Biology students given peer education by university students 54**								
Manchester	M1	21	21	100	19	90	15	71
Cardiff A	A1	26	26	100	24	92	19	73
Cardiff B	B1	7	7	100	7	100	6	86
Total biology students		54	54	100	50	93	40	74
**Non-biology students receiving peer education from biology students 127**								
Manchester	No peer education by Manchester biology students							
Cardiff A	A2	11	11	100	10	91	11	100
Cardiff B	B2	15	3	20	3	20	3	20
	B3	22	10	45	8	36	5	23
	B4	16	5	31	5	31	4	25
	B5	24	18	75	17	71	12	50
	B6	15	9	60	9	60	8	53
Totals where data collected		103	56	54	52	50	38	37
Cardiff B no questionnaires	B7 *	24	0*	-	-	-	-	-
Overall Totals for non-biology students		127	56	44	52	41	38	30
**Control schools 120 biology students (Cardiff 95, Manchester 25) invited to participate**								
Manchester		25	18	72	N/A	N/A	21	84
Cardiff		50	50	100	N/A	N/A	45	90
Total control students		75	68	91	N/A	N/A	66	88

* Students were too disruptive to allow for any data collection.

**Table 2 antibiotics-09-00146-t002:** Reported antibiotics prescribed in control and intervention students responding to any text versus GP data and students responding to all three-monthly texts versus GP data.

Antibiotic Use Reported by SMS, GP Data, and Questionnaire Data	Antibiotic Prescribed at GP Surgery		% Agreement (kappa)	
	Response	No	Yes	
Antibiotic use reported by SMS in students responding to any SMS versus GP data (35)	No *N*= 28	27	1	91% (0.72)
	Yes *N* = 7 (20%)	2	5	
	Total	29	6	86% (0.67)
Antibiotic use reported by SMS in students responding to all three-monthly texts versus GP data (22)	No *N* = 15	14	1	
	Yes *N* = 7 (32%)	2	5	
	Total	16	6	
Antibiotic use reported by student in 3-month questionnaire (25)	No *N* = 16	15	1	64% (0.06)
	Yes *N* = 9 (36%)	8	1	
	Total	23	2	

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
