# Peer review of "Peer-Education as a Tool to Educate on Antibiotics, Resistance and Use in 16–18-Year-Olds: A Feasibility Study"

_antibiotics, 2020, doi:10.3390/antibiotics9040146_

Round 1
Reviewer 1 Report
The paper is well conceived and well written.
No further remarks.
Author Response
Thank you for your comments, pleased to hear you enjoyed reading it.
Reviewer 2 Report
This article describes a feasibility study using university students and biology students as peer educators using the e-Bug curriculum to teach teens ages 16-18 about antibiotic use. This study is very interesting, and the tables/figures are very helpful for the reader. There are some minor issues I would recommend clarifying before being published.
Introduction, page 2, lines 47-49: Please provide the reference for this significant increase in knowledge/improved behavior, confidence, and communication skills.
Methods: Why did you choose the number of schools you did as your “convenience sample”?
Methods: Why didn’t you have university students educate non-biology students?
Results, section 2.4, lines 147-148: Did you combine antibiotic use data for all data collection methods or just for the GP data? It seems like to determine if your intervention really worked, you would need to know if the students who received the PE had a lower antibiotic usage than those who did not receive the PE. Why wasn’t this analysis done?
Author Response
Thank you for your time in reviewing this manuscript. Please see the attachment.

Reviewer 3 Report
1. Evaluation data from this study clearly indicate that delivering e-Bug Peer-Education (PE) for antibiotic information for 16-18 year old students in this school setting is feasible with the supportive school teaching staff.
2. The study data show that single tiered PE by University students appeared to be easier to regulate and manage in a setting where students have time constraints, indicating that single tiered PE training by University students may be easier to regulate and manage in a setting where students and teachers have time constraints.
3. The authors conclude that SMS collection of antibiotic data information is easier to conduct and has similar accuracy compared to GP collected data.
Author Response

(The authors gave the same response as above.)
